# Dual-Energy and Photon-Counting Computed Tomography in Vascular Applications—Technical Background and Post-Processing Techniques

**DOI:** 10.3390/diagnostics14121223

**Published:** 2024-06-11

**Authors:** Marcin Stański, Ilona Michałowska, Adam Lemanowicz, Katarzyna Karmelita-Katulska, Przemysław Ratajczak, Agata Sławińska, Zbigniew Serafin

**Affiliations:** 1Department of General Radiology and Neuroradiology, Poznan University of Medical Sciences, 61-701 Poznań, Poland; kkatulska@ump.edu.pl; 2Department of Radiology, National Institute of Cardiology, 04-628 Warsaw, Poland; imichalowska@ikard.pl; 3Department of Radiology and Diagnostic Imaging, Nicolaus Copernicus University, Collegium Medicum, 85-067 Bydgoszcz, Poland; adam.lemanowicz@cm.umk.pl (A.L.); p.ratajczak@cm.umk.pl (P.R.); agataslawinska@cm.umk.pl (A.S.); serafin@cm.umk.pl (Z.S.)

**Keywords:** diagnostic imaging, computed tomography, dual-energy computed-tomography, photon-counting computed-tomography

## Abstract

The field of computed tomography (CT), which is a basic diagnostic tool in clinical practice, has recently undergone rapid technological advances. These include the evolution of dual-energy CT (DECT) and development of photon-counting computed tomography (PCCT). DECT enables the acquisition of CT images at two different energy spectra, which allows for the differentiation of certain materials, mainly calcium and iodine. PCCT is a recent technology that enables a scanner to quantify the energy of each photon gathered by the detector. This method gives the possibility to decrease the radiation dose and increase the spatial and temporal resolutions of scans. Both of these techniques have found a wide range of applications in radiology, including vascular studies. In this narrative review, the authors present the principles of DECT and PCCT, outline their advantages and drawbacks, and briefly discuss the application of these methods in vascular radiology.

## 1. Introduction

DECT is a technology that already has a long history. It was Sir Godfrey Hounsfield in the 70s who found a way to differentiate calcium and iodine using two energy spectra from X-ray beams [1]. This method is based on the knowledge of distinct atomic numbers and unique k-edge characteristics of various materials. These qualities are crucial because they define the different contributions of Compton scattering and the photoelectric effect in X-ray attenuation.

At the beginning of the 80s, DECT was applied mainly in bone densitometry, as in Somatom DR (Siemens Healthcare, Erlangen, Germany). This device used fast tube potential switching for acquiring two-photon energy spectra. The usage of this method beyond densitometry was limited because of the low computational power of hardware. However, the XXI century brought a breakthrough in the clinical application of this method, as the abilities of processors evolved rapidly. First, scanners with two sources (Somatom Definition DS, 2006; Somatom Definition Flash, 2009; Siemens Healthineers, Erlangen, Germany) and multilayer detectors (Brilliance-64; Philips Medical Systems, Bothell, WA, USA, 2015) were introduced and the method of fast tube potential switching was developed further (Revolution GSI, Discovery 750 HD; General Electric Healthcare, Milwaukee, WI, USA, 2010) [2].

DECT enables obtaining and reconstructing a broad range of images. Kilovolt peak images are similar to those from SECT, simulating single energy spectra. They are available for dual-layer, rapid kVp-switching, and split-filter DECT. Dual-source DECT enables obtaining kilovolt peak images for kilovolt peak pair or kilovolt peak-equivalent images. The latter are a weighted average of data obtained with two peaks. These data constitute a reconstruction that appears similar to images obtained with one, chosen by the user, kilovolt peak value. VMCs are simulations of scans obtained with photons with a single energy level. Their application is especially valuable, as the attenuation of iodine increases when the simulated photon energy level is low. Another technique is material decomposition, which relies on different levels of participation of Compton scattering and the photoelectric effect in the X-ray attenuation of various materials. This allows for obtaining images with iodine removed or accentuated, and with urine and calcium removed [3,4].

Photon-counting computed tomography (PCCT) represents a cutting-edge technological advancement in the realm of energy-resolving, direct-conversion X-ray detectors. Following an extensive period of research and development spanning 15 years, this technology has only recently been incorporated into clinical CT equipment. The underlying principles of PCCT diverge significantly from those of conventional CT detectors. These conventional detectors are EIDs, which create signals proportional to the total energy of photons collected during one measurement interval. In PCCT, PCDs are used, which directly convert the energy of single photons into electrical signals. The device includes only electrical pulses with heights above the thresholds characteristic of noise [5]. Thus, this method allows for tremendously decreasing the level of electrical noise and increasing the SNR. Moreover, it can also be used in double-energy imaging [2]. The emergence of PCCT holds the potential to revolutionize the clinical CT landscape by capitalizing on its numerous inherent benefits and addressing several limitations inherent in the current state-of-the-art CT systems.

The primary objective of this article is to briefly discuss the principles of DECT and PCCT, outline its advantages over standard single-energy computed tomography (SECT).

## 2. Techniques of DECT

### 2.1. Rapid-kVp Switching DECT

The first method of dual-energy acquisition, employed since the 1980s, involved using an X-ray tube that rapidly alternated between high and low tube voltages—a technique known as rapid-switching kilovoltage–peak dual-energy CT [6]. This laid the foundation for the commercially used fast-kVp switching technique. It is used by the device produced by General Electric Heathcare, Revolution GSI, Discovery 750 HD. In this system, an X-ray tube quickly alternates between 80 and 140 kVp, along with an ultra-fast registering detector. In this way, two data sets are immediately obtained. Material decomposition algorithms can be implemented using either reconstructed images or projection data, aiding in artifact reduction within VMCs [2]. However, this technique has some drawbacks, including the absence of independent tube filtration for optimizing spectral separation, fixed settings of 80 and 140 kVp causing low-energy photon starvation in obese patients, and the unavailability of tube current modulation for radiation dose reduction [7]. A scheme of the rapid-kVp switching system is presented in Figure 1.

### 2.2. Dual-Source DECT

This type of scanner was introduced by Siemens Healthineers (Somatom Definition Flash, Somatom Force). Two tubes and two sets of detector rings are built-in. The tubes are positioned at 90° to themselves and are sources of distinct, independent X-ray beams of low (70–80 kVp) and high (140–150 kVp) energy spectra. The latest, third generation of dual-source systems have a tin filter absorbing lower-energy photons in front of the high-peak kilovoltage tube. This allows for an increase in mean spectral energy, improved spectral separation of the two tube energies, and increased CNR in material-specific images [7]. The ability to optimize the spectral filtration independently for each tube–detector pair is the main advantage of the dual-source system compared with other systems. The main disadvantage of this system is the difference between the FOVs of the high- and low-energy tubes. The disparity between the sizes of the FOVs of the tubes decreases with each new generation of the equipment; in the current third generation, the FOV of the low-energy tube is 50 cm, whereas the FOV of the high-energy tube is 35 cm. This limits the evaluation of peripherally located structures. The simultaneous use of both X-ray sources causes scattered radiation to be detected by the detector of the second tube, which requires a scatter correction algorithm [2]. Additionally, the shift of the tubes causes a minimal delay between the two lamps that generates misregistration artifacts. The scheme of the dual-source system is presented in Figure 2.

### 2.3. Split-Filter DECT

Owing to the split filter system, there is a possibility to obtain both high- and low-energy images. This technology is used in two devices by Siemens Healthineers, namely Somatom Definiton Edge and Somatom go.Top. In these machines, two filters, made of metals—gold and tin, are used between a lamp and a patient. In this way, the device can provide two beams of X-rays of different energy spectra—120 kV and 80 kV [3]. The advantage of this solution is the lower cost of the equipment, associated only with an additional set of tube filters. The main weakness of this system is the limited spectral separation, and as a result, decreased CNR. We provide an image presenting the split system in Figure 3.

### 2.4. Multilayer Detector CT

This system utilizes layered detectors, which collect data from a single X-ray tube of 120 kVp [3]. The more superficial layer is made of yttrium and obtains low-energy data, whereas the deep layer is made of gadolinium–oxysulfide and collects high-energy data. The significant advantages of this system include simultaneous data collection and the continuous acquisition of two datasets, meaning that every examination can be analyzed using dual energy. Additionally, this system does not increase the patient’s radiation exposure. Compared with a fast-kVp switching system, the tube rotation time is shorter, allowing for higher temporal resolution. When compared with a dual-source system, the main benefits are the absence of field-of-view limitations and cross-scatter effects. However, this system does have some downsides, such as the relatively high overlap of energy spectra and the potential for miscalculations in monoenergetic images [3]. A scheme of this system is presented in Figure 4. An example of such a device is IQon Spectral CT by Phillips Medical System.

## 3. Photon-Counting CT

### Technical Background

The method of PCCT is mainly based on a specific kind of X-ray detector. EIDs, which are used classically in CT devices, are composed of two layers. The more superficial layer is built of ceramic scintillators, whereas the deeper one is made of a photodiode array. Ceramic scintillators convert X-ray photons into secondary visible-spectrum photons, which are absorbed by a photodiode array, generating an electrical signal proportional to the total energy. This signal is amplified, converted to a digital format, and processed for image reconstruction. However, the energy information of individual X-ray photons is lost. Moreover, in EIDs, there are included reflective layers (septa). They cause “dead space” on EID surface, limiting geometric dose efficiency [5].

PCDs utilize a direct conversion technique, coupling a high atomic number semiconductor sensor with a readout circuit. X-ray photons are then directly translated into electrical signals. This allows for reducing noise, improving resolution, lowering the dose, and reducing dark current. Moreover, the electric pulse height is proportional to the deposited energy, allowing for X-ray photons to be sorted into energy bins using multiple electronic comparators [8]. Figure 5 and Figure 6 present schematic drawings of conventional and photon-counting detectors.

The introduction of PCD CT systems marks a significant milestone in the development of medical CT scanners following the advent of multi-row and DECT systems. Compared with conventional CT systems, PCD CT systems carry many inherent advantages and have the potential to overcome the limitations of previous CT systems. PCD CT systems can measure and quantify the energy of each photon and generate a wide range of spectral reconstructions [9]. Despite their recent introduction, PCDs have already proven their superiority over devices with classic detectors, with better sharpened contrast and spatial resolution, spectral reconstructions, dose efficiency, and noise reduction [8]. Some of these advantages have been already proven in post-EVAR surveillance [10].

## 4. Postprocessing Techniques

A brief summary of the different names of postprocessing techniques, depending on vendors, is provided in Table 1.

### 4.1. Material Decomposition

#### 4.1.1. DECT

The ability of DECT to decompose materials into their elemental components is based on their different features, which impact X-ray attenuation. In the range of energies used in radiological examinations (E < 150 keV), these attenuation profiles are mainly influenced by the photoelectric effect and Compton scattering processes. The attenuation coefficients of elements with K- and L-edges outside the diagnostic energy range can be modeled as a linear combination of these aforementioned phenomena. This allows for obtaining atomic numbers (Z), attenuation profiles, and material mass density maps [2,11]. Thus, various types of reconstructions, which inform about the concentrations of certain elements and substances in the material, may be produced. Depending on the type of scanner, it is possible to apply two- and three-material decomposition algorithms. The two-material decomposition algorithm, available in single-source systems (like rapid kVp switching and dual-layer systems), allows for the differentiation of two selected materials, typically water and iodine. Three-material decomposition algorithms are used in dual-source and split-filter systems, and depending on the type of examination, they allow for the simultaneous differentiation of, for example, water, calcium, and iodine in angiographic studies, or soft tissue, iodine, and fat in abdominal CT scans.

#### 4.1.2. PCCT

Improved material decomposition is possible in PCCT as photons are sorted into different energy bins [12]. Thus, the selection of energy thresholds may be tailored toward specific materials one wants to identify, such as iodine. An example of iodine maps generated by PCCT, which are helpful in the diagnostics of pulmonary thrombosis, is provided in Figure 7. This ability of PCCT is still under study and, although it seems to be more robust than that in DECT, it is not unlimited. As already mentioned, in DECT, two measurements at different energies are taken to differentiate two materials, typically water and iodine. Any other materials may be then described as a combination of these two, and further measurements with different energy levels usually do not provide new information. However, they are relevant in the presence of a material with K-edges, such as gadolinium or certain metals (gold and tungsten), as its relation between energy and X-ray attenuation is different from that of structures occurring naturally in the human body [9]. Therefore, improved metal decomposition of PCCT may be used mainly in applications such as dual-contrast agent imaging or reducing metal artifact burdens [13].

### 4.2. Virtual Non-Contrast

#### 4.2.1. DECT

VNC reconstructions are images created by the subtraction of iodine [14]. Numerous studies have proven that VNC images can substitute for TNC images with sufficient quality [15,16,17,18]. Among the most common utilizations of the VNC technique are those related to vascular CT examinations. VNC imaging has proven useful in calcium scoring in coronary CT angiography [19]. Iodine quantification obtained through VNC images can greatly increase diagnostic accuracy in differentiating malignant and bland thrombus in portal vein thrombosis [20]. VNC images have proven to be reliable in settings of the endoleak assessment in patients after EVAR, with possible radiation dose reduction [14,21]. However, VNC images have been proven prone to the iodine content. This results in differences in attenuation between TNC and VNC reconstructions obtained from various phases of examination [14,22]. Figure 8 presents the differences between TNC and two post-contrast VNC images.

Material decomposition and VNC techniques can be used to obtain iodine maps. The extracted iodine can also be used to construct precise maps, presenting the iodine distribution in the tissues [23]. Moreover, color-coding of iodine, highlighting the iodine content on gray-scale VNC images, can be performed. Such reconstructions have been found useful, particularly in oncological applications, but also in endoleak detection [20,24,25].

Scans are of the same patient, the same axial slice, and with the same window (W 400, L40).

#### 4.2.2. PCCT

As with most DECT reconstructions, VNC images may be also obtained using PCCT. They have been shown to be particularly valuable in post-EVAR assessment. In the study by Decker et al. [10], VNCpc demonstrated similar value in follow-up scans after EVAR as TNC images. Interestingly, the quality of reconstructions was higher than that of those created by conventional DECT. They had a higher SNR and smaller differences with TNC scans. In the study by Mergen et al. [26], the diagnostic quality of VNC images was reached in 99–100% of patients, although radiologists rated their subjective quality lower than that of TNC images. Importantly, the authors found no significant difference between VNC reconstructions obtained from different phases of examination. There are also other studies presenting similar results [26,27]. However, in some papers, significant differences in CT numbers with TNC were found [28], which suggests a strong need for further research.

Figure 9 is an example of the high resolution and wide dynamic range of VNC images obtained with PCCT—in this case, it enabled the precise assessment of calcifications in the patient’s aortal valve.

### 4.3. Virtual Non Calcium (VNCa)

#### 4.3.1. DECT

Advances in DECT techniques have led to the introduction of novel three-material differentiation algorithms. They remove calcified plaques with no changes in adjacent tissues and iodine contrast in vessel lumens [29,30]. Previously, widely used calcium subtraction methods relied on the application of fixed CT value thresholds for the removal of calcifications. VNCa algorithms, similar to VNC images, estimate the amount of calcium in the datasets and subtract it. VNCa algorithms can help to assess vessel stenosis caused by calcified plaques. Blooming artifacts may cause the overestimation of vascular stenosis, potentially leading to unnecessary vascular interventions [31]. Furthermore, dense plaques may cause streak and beam hardening artifacts, obscuring the vessel’s lumen and surrounding soft tissues [30,32]. VNCa algorithms have already proven their value in assessing carotid artery stenosis, mitigating the problem of the overestimation of stenosis on conventional CTA images [30].

#### 4.3.2. PCCT

The evaluation of vessels in the presence of extensive calcification may be further improved with PCCT. The algorithms, similarly to those used in DECT, are based on iodine-calcium decomposition [33]. The VSPP images are created, which preserve the information from different energy bins and enable calcium and iodine separation [34]. PCCT is especially promising in limiting the stenosis overestimation caused by calcium in vessel walls, both in larger and smaller vessels [33]. Allmendinger et al. evaluated the usefulness of the PCCT PureLumen Algorithm, developed by Siemens Healthineers, in coronary artery assessment using a vascular phantom. They found that it decreased blooming artifacts in strongly calcified vessels and increased image interpretability, with high interrater agreement. Interestingly, the quality of images remained high over a wide range of heart rates [35]. Not only did PCCT show a potential to improve coronal stenoses quantification in phantoms, but also in vivo, in the first clinical experience in patients with a high calcium score [33,34]. The algorithms described in this paragraph also have robust performance in visualizing the lumen of stented coronary arteries, as demonstrated in Figure 10.

### 4.4. Virtual Monoenergetic Images

#### 4.4.1. DECT

VMIs are reconstructions that simulate images obtained with a beam of photons with a single energy. They can be obtained in the projection space or image space. The available levels of energy (in keV) vary between the producers (usually they range between 40 keV and 200 keV). These reconstructions are valuable tools in the interpretation of contrast-enhanced images. Low-keV images (40–70 keV) have increased iodine contrast, as these levels of energy are closer to the k-shell binding energy of iodine (33.2 keV) [36], but their downside is the decreased CNR [37]. This can be advantageous when the lesion enhances slightly, as low-energy VMIs are better than conventional CT, visualizing subtle enhancements [38]. Low-keV data sets can be particularly useful for improving iodine contrast [39], for example, in small contrast volumes at slow injection rates [7], improved detection and delineation of poorly-enhancing lesions [40,41], endoleak detection [42,43], stent visualization [39,44], assessment of the coronary vasculature, and functional evaluation of the myocardium [45,46]. Figure 11 presents sample endoleaks in conventional reconstruction and low-keV VMI.

On the other hand, high-keV images (100–150 keV) reduce image noise, but provide a lower contrast-to-noise ratio. They allow for the reduction of artifacts related to hyperdense structures, such as metal stents and calcified atherosclerotic plaques [47]. In the phantom study by Mangold et al., high-energy VMIs were shown to be beneficial in assessing in-stent restenosis in coronary arteries CT [48]. Furthermore, VMIs at 140 keV significantly improve the accuracy of in-stent luminal diameter measurements [49]. High-energy keV images are also helpful in assessing regions with artifacts due to coils or metal clips, as demonstrated by Winkhlofer et al. [50]. However, the combination of high keV and MAR may severely hamper the visualization of poorly enhanced lesions and stents (Figure 12) [39].

#### 4.4.2. PCCT

As already mentioned, VMI reconstructions have a wide range of applications in vascular imaging [39]. There is hope for further improvement in this field as PCCT scanners develop. In DECT, a scanner obtains two sources of information (e.g., by dual X-ray tubes with different voltages); a PCCT scanner can gain even more information by establishing more than two energy thresholds during the separation of detected photons [51]. The VMIs produced by PCCT scanners have been extensively studied as they are expected to have improved spatial resolution and lower image noise with no need for a higher radiation dose.

Many studies found that PCCT produces higher-quality VMIs than DECT scanners at low radiation doses [52,53]. Vrbaski et al. [51] reported that PCCTs have comparable accuracy in VMI imaging between low and standard radiation dose levels, whether or not the DECT performance was dependent on the radiation dose. Liu et al. [54] found that VMIs obtained with PECT in a cardiac phantom for coronary CT with different heart rates had especially high spatial and temporal resolutions with a low applied radiation dose.

On the other hand, a phantom study by Boji et al. [55] demonstrated better performance of PCCT than DECT in estimating VMI only at levels below 60 keV. Moreover, Sartoretti et al. [56] found no difference in estimating VMI between PCCT and DECT scanners.

To sum up, although the comparison of PCCT with DECT in VMI reconstruction remains a matter of research, its main advantage seems to be better or equal image quality with an equal or lower radiation dose.

## 5. Limitations

We did not present the pitfalls in the interpretation of DECT and PCCT images, as it is beyond the scope of this mini-review. Pitfalls are characteristic of each type of the abovementioned reconstructions. For further reading, we could recommend the comprehensive review on the pitfalls by Parakh et al. [4].

## 6. Conclusions

DECT and PCCT are emerging technologies that offer additional information, which is not accessible through conventional CT. They open new horizons in imaging, and their application in clinical settings is still broadening. Possible applications are demonstrated in Table 2.

The application of spectral CT systems in vascular applications allows for an enhancement in the diagnostic value of examinations. Although invasive angiography remains the gold standard, especially in the assessment of coronary artery disease, it would be beneficial as a non-invasive alternative [57]. The most useful reconstructions are those obtained with material decomposition and VMI reconstructions. Owing to the use of virtual non-enhanced phases, spectral CT angiography can be performed with a significantly lower effective radiation dose and potentially reduced contrast agent dosage. VMI reconstructions enhance the visualization of vessels and may assist in the evaluation of images marked by metal artifacts. The advancements in DECT and PCCT technologies and their growing applications in various clinical settings demonstrate their potential to revolutionize the field of medical imaging and improve patient outcomes.

## Figures and Tables

**Figure 1 diagnostics-14-01223-f001:**
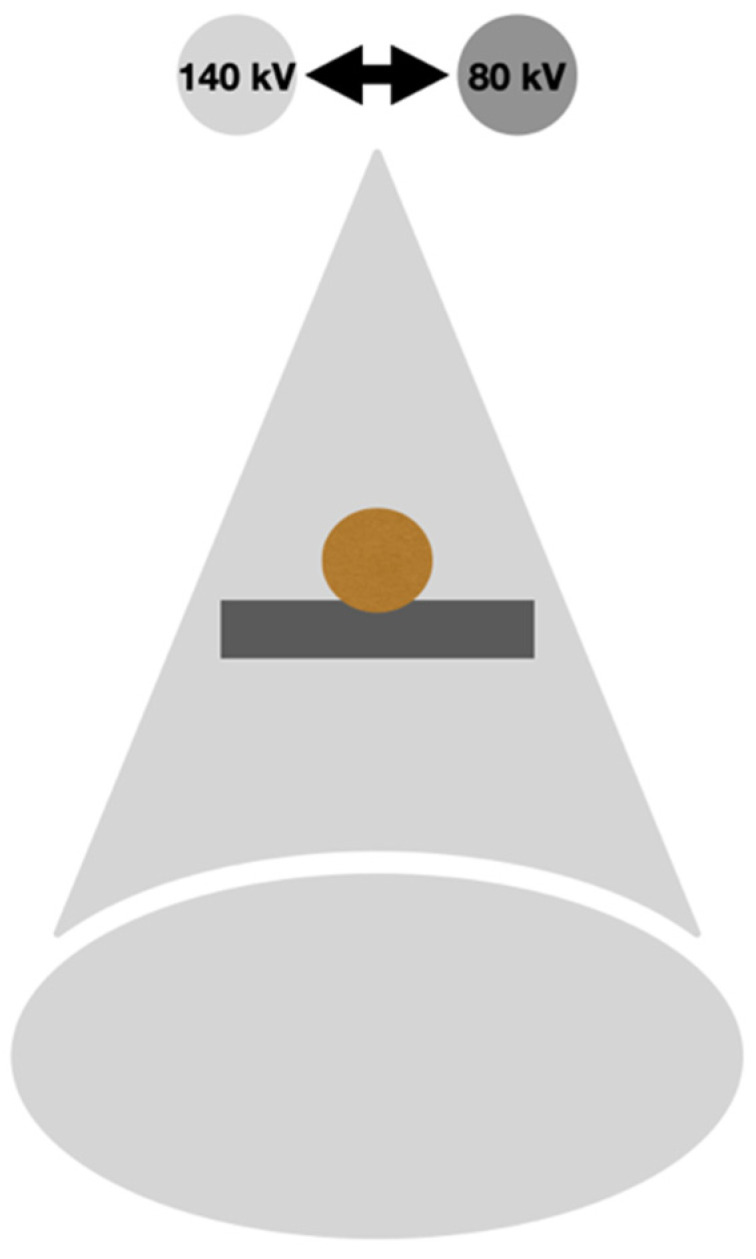
Scheme of rapid-switching DECT.

**Figure 2 diagnostics-14-01223-f002:**
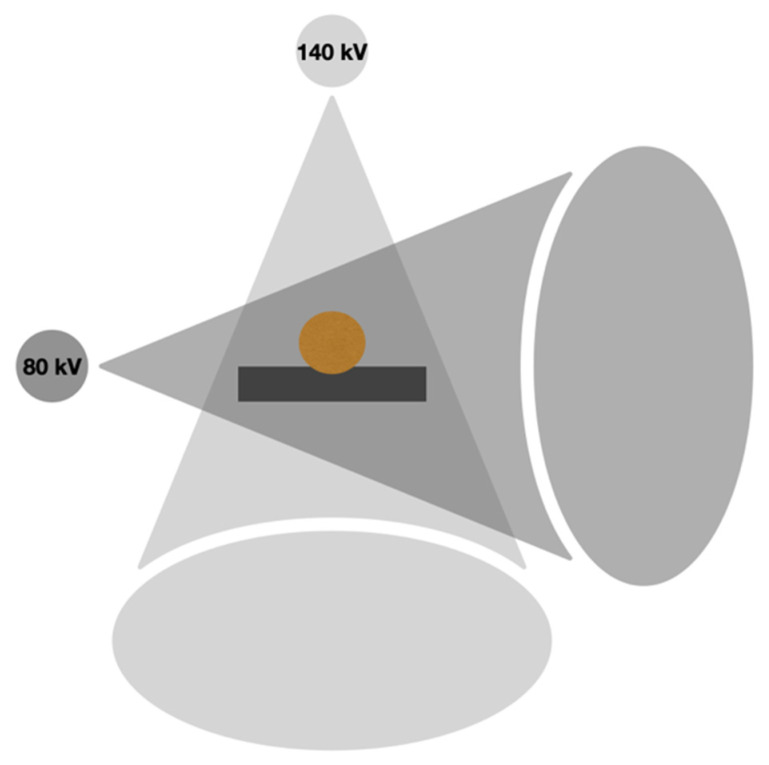
Scheme of dual source DECT.

**Figure 3 diagnostics-14-01223-f003:**
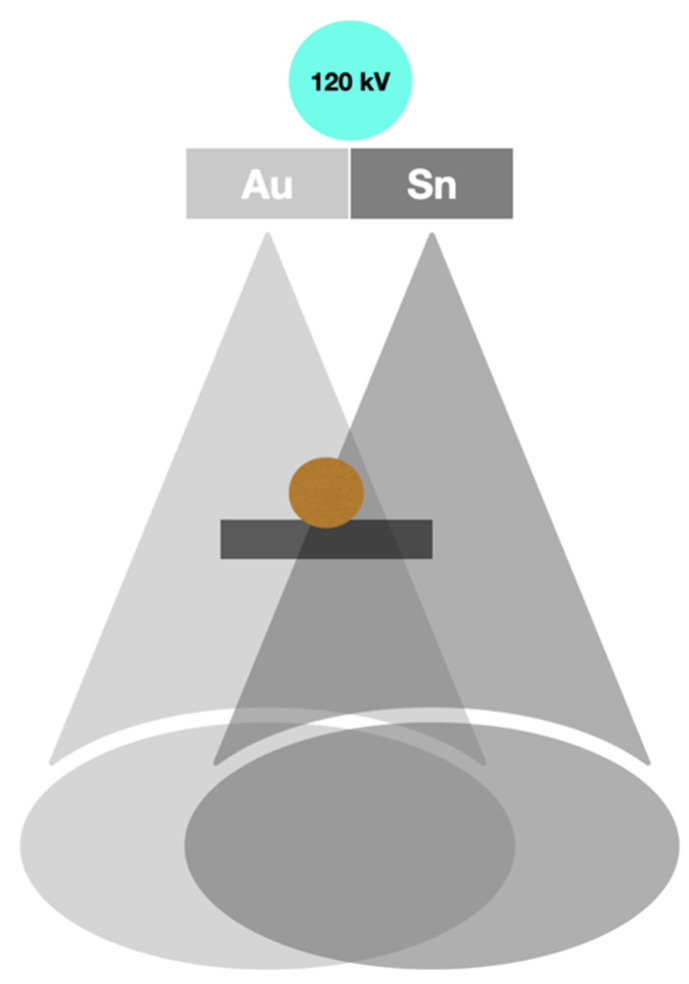
Scheme of split-filter DECT.

**Figure 4 diagnostics-14-01223-f004:**
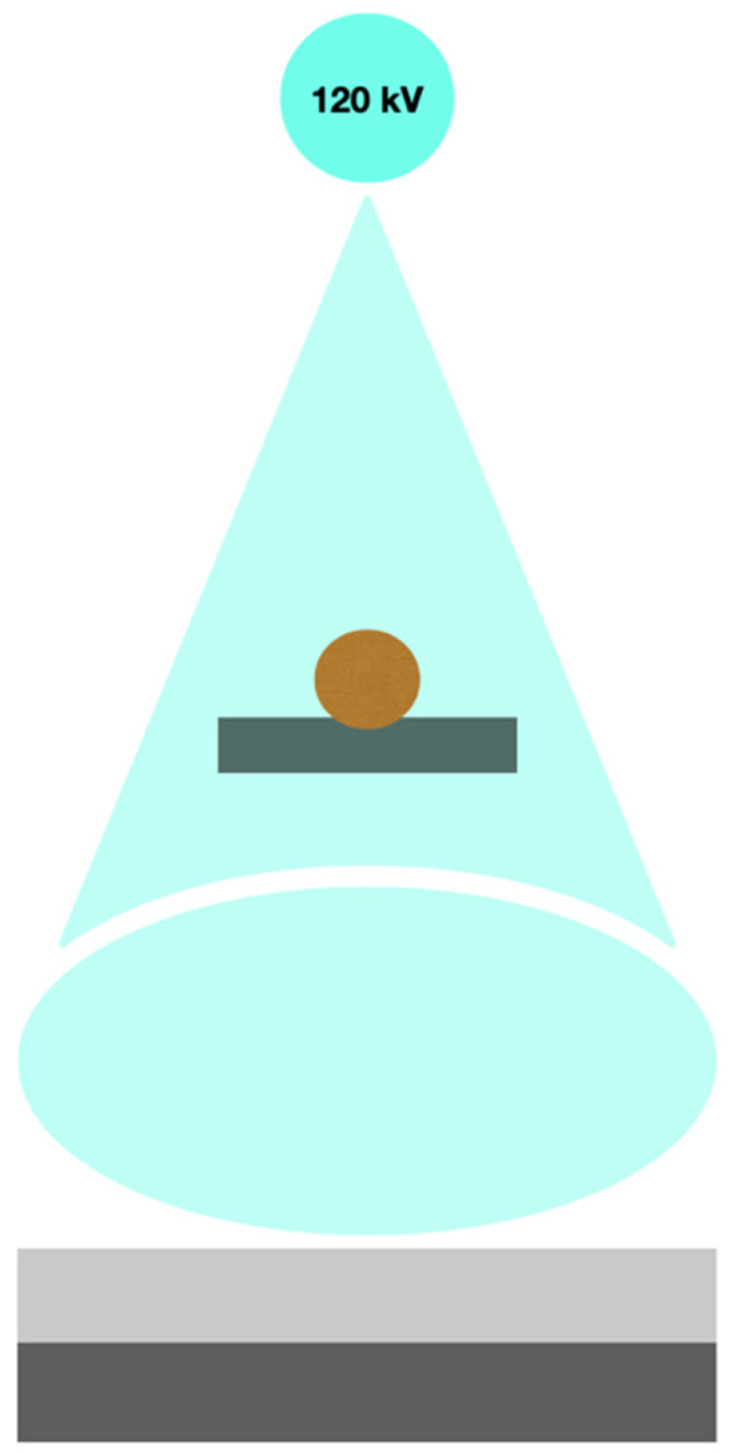
Scheme of multilayer detector DECT.

**Figure 5 diagnostics-14-01223-f005:**
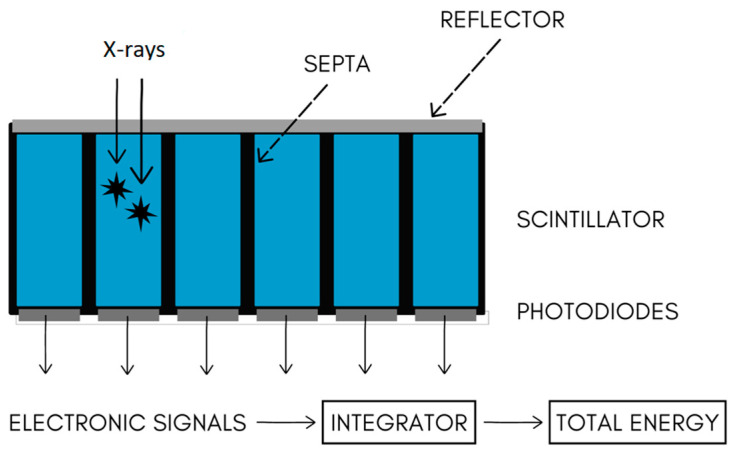
Scheme of conventional CT detector.

**Figure 6 diagnostics-14-01223-f006:**
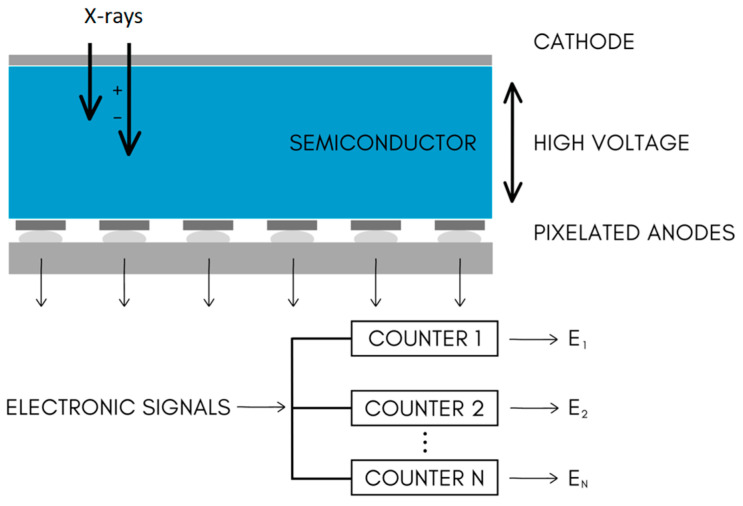
Scheme of photon-counting CT detector.

**Figure 7 diagnostics-14-01223-f007:**
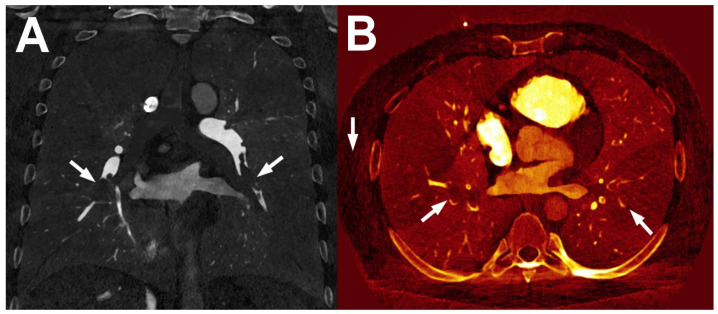
Presentation of pulmonary thrombosis (arrows) in PCCT iodine maps on a gray-scale image (**A**) and a colorful reconstruction (**B**).

**Figure 8 diagnostics-14-01223-f008:**
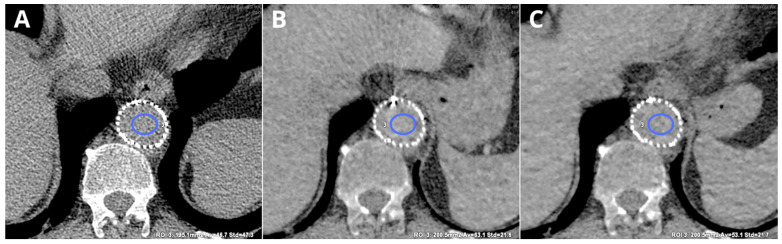
Differences in attenuation in aorta in TNC and VNC images obtained from different phases of examination. TNC ((**A**)—46.7 ± 47.3 HU), VNCa ((**B**)—63.1 ± 21.6 HU), and VNCd ((**C**)—53.1 ± 21.7 HU).

**Figure 9 diagnostics-14-01223-f009:**
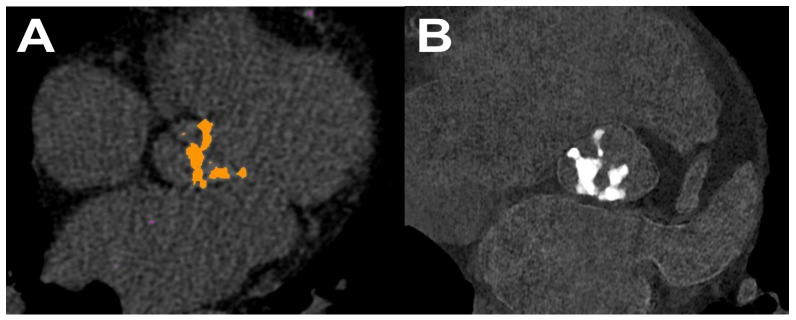
Imaging of aortic valve calcifications at PCCT. Conventional calcium scoring (**A**). Evaluation of calcifications’ morphology (**B**). Note significant differences in calcium density.

**Figure 10 diagnostics-14-01223-f010:**
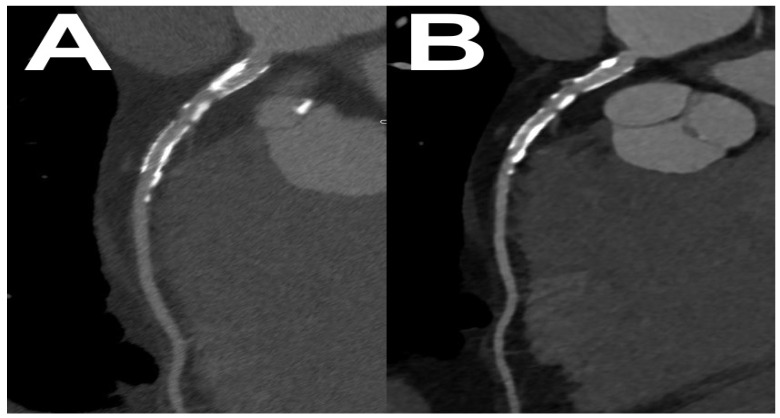
Imaging of a LAD stent at PCCT using 0.2 mm (**A**) and 0.4 mm (**B**) slice thicknesses.

**Figure 11 diagnostics-14-01223-f011:**
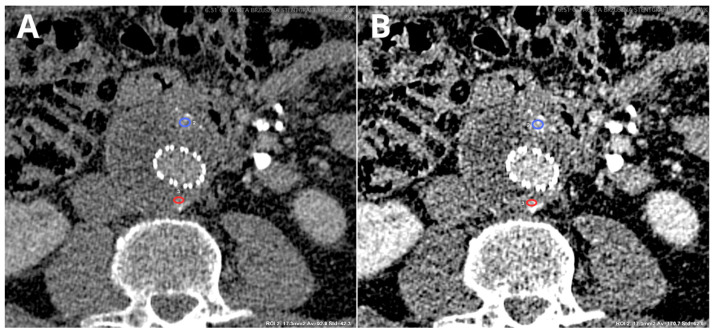
Images presenting differences in CT numbers (average ± SD) in two low-flow type II endoleaks associated with the inferior mesenteric artery (blue circle) and one of the lumbar arteries (red circle) in standard, linearly blended (LB), and 50 keV VMI reconstructions. Sixty-second delayed phase of CT examination. Reconstructions: linearly blended ((**A**)—blue circle 92.9 ± 42.3 HU, red circle 86.7 ± 39.2 HU), 50 keV VMI ((**B**)—blue circle, 170.7 ± 62.6 HU, red circle, 165.9 ± 59.9 HU). Same axial slice, same patient. Images presented with the same window settings (W 500, L 100).

**Figure 12 diagnostics-14-01223-f012:**
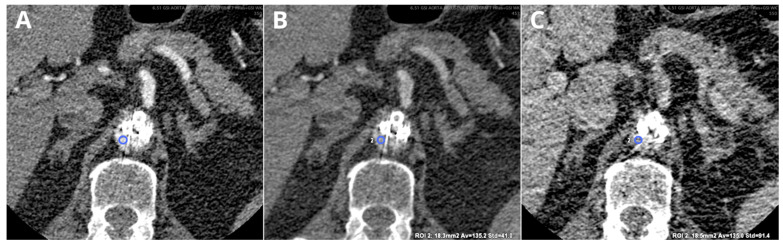
Comparison of linearly blended (LB) (**A**), 130 keV VMI (**B**), and 130 keV MAR reconstructions (**C**). Type III endoleak in patient after fenestrated EVAR, arterial phase of CT examination. Reduced noise level (SD) in 130 keV VMI MAR images ((**B**)—blue circle, 135.2 ± 41.0 HU) compared with LB reconstruction ((**A**)—244.5 ± 58.9 HU). Complete obscuration of stents lumen, aorta, and endoleak on MAR images ((**C**)—blue circle, 135.0 ± 94.4 HU). Images presented with the same window settings (W 500, L 100).

**Table 1 diagnostics-14-01223-t001:** Image types in different dual-energy CT platforms. DECT—dual-energy CT [4].

	Siemens Healthineers	GE Healthcare	Philips Healthcare
Technique			
DECT system	Dual source, split filter	Rapid kVp switching	Dual layer
Material decomposition images—iodine	Liver non-calcium, lung perfusion, brain hemorrhage, heart, pulmonary blood volume	Iodine (water), iodine (calcium), iodine (hydroxyapatite), iodine (fat)	Iodine no water, iodine density
Material decomposition images—calcium	Brain hemorrhage, kidney stones	Calcium (water), calcium (iodine), calcium (uric acid),	Uric acid, uric acid removed
Material decomposition—urate	Kidney stones, gout	uric acid (calcium), uric acid (hydroxyapatite)	Uric acid, uric acid removed
Virtual non-contrast images	Liver virtual non-calcium, virtual unenhanced	Water (iodine), virtual unenhanced	Virtual unenhanced, iodine removed
Virtual non-calcium image	Bone removal, hard plaque, brain hemorrhage, blood marrow	Iodine (calcium)	Calcium suppressed
Virtual kilo-electronovolt images	Virtual monoergetic images	Virtual monochromatic images	Monoenergetic Images

**Table 2 diagnostics-14-01223-t002:** Summary of DECT and PCCT advantages over conventional SECT and their vascular applications.

**Reconstruction Technique**	**Advantages**	**Applications**
Material decomposition	Decompose materials into their elemental components	- Separation of calcium from iodine
Virtual non-contrast	Non-contrast scan can be omitted	- Calcium scoring - Endoleak assessment - Iodine quantification in tissues
Virtual non-calcium	Robust calcium subtraction method	- Assessing carotid artery stenosis, - Reducing blooming artifacts - Mitigating the problem of overestimation of stenosis on conventional CTA images
Virtual monoenergetic images	Mimic the attenuation values of an image obtained with single energy	- Improving iodine contrast - Improved detection and delineation of poorly-enhancing lesions - Endoleak detection, stent visualization - Assessment of the coronary vasculature and functional evaluation of the myocardium

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
