# Peer review of "Dual-Energy and Photon-Counting Computed Tomography in Vascular Applications—Technical Background and Post-Processing Techniques"

_diagnostics, 2024, doi:10.3390/diagnostics14121223_

Round 1

Reviewer 1 Report

Comments and Suggestions for Authors

Dear Authors, 

There are numerous strengths in this study including current and novel DECT and PCTT  application. I found this study makes a valuable contribution to the literature. Because the CT technical evolution and innovations, but I suggest you improve and detail DECT and PCCT characteristics.

I would add in the introduction a short summary of the technical step, as you presented the various CT scanners, it appears as they are contemporary instead they represent different step of technical improvements and summarize the spectrum of images available with DECT. 

Another suggestion is to briefly explain the principal concept before introducing the scanners (kilovolt peak images and equivalent, monochromatic and material density).

I did not understand the reason why you mentioned some of the commercially available platforms and not all, the nomenclature also of the different scanners may be different and a summary table that compare them is missing. 

Pitfalls should be added.

Author Response

Dear Reviewer, 

Thank you for considering our paper. We are very grateful for your suggestions.

Here are our answers  :

1, 2. We have added a short summary on the technical step, concepts and types of images 

3. We have added a summary table which introduces the nomenclature of different vendors 

4. Since pitfalls are beyond the scope of the review, we have given a reference to a comprehensive review of pitfalls, for an interested reader. 

We have marked changes in text with yellow color. 

Thank you once again,

Yours sincerely 

the Authors

Reviewer 2 Report

Comments and Suggestions for Authors

Interesting review on advanced CT techniques in vascular diagnosis. Following are my few suggestions to improve the manuscript.

Lines 147-8, “Some of these advantages have been already proven in post-EVAR surveillance”: explain the meaning of the acronym, and give an appropriate reference.

Line 306, “Figure 10. Imaging of a LAD stent at …”: explain the meaning of the acronym.

Since there are many technical acronyms, I suggest the authors to provide a glossary at the beginning (or at the end) of your manuscript. So avoiding repeating the meaning of each acronym, saving space in the paper, and helping the reader in the comprehension. For instance:

-line 201, “…virtual noncontrast (VNC)…”: the meaning of the acronym has been already introduced in line 179.

-line 267, “4.3. Virtual non calcium (VNCa)”: the meaning of the acronym has been already introduced in line 179.

-line 271, “Advances in Dual-Energy CT (DECT) techniques have led to the introduction of novel …”: the meaning of the acronym has been already introduced in line 27.

-lines 275-6, “Virtual Non-Calcium (VNCa) algorithms …”: the meaning of the acronym has been already introduced in line 179.

- line 313, “Virtual monoenergetic images can be reconstructed from dual-energy CT …”: use the acronym “DECT” already introduced in line 27.

Is angiography still the gold-standard of comparison when dealing with DECT and PCCT? Please, comment on it in the (Discussion)/Conclusions section, or somewhere in the manuscript.

Author Response

Dear Reviewer,

Thank you for considering our paper and helpful suggestions. 

Here are our answers :

  1. We have added a glossary at the beginning of the paper, as you suggested
  2. We have mentioned angiography as a golden standard - we feel that it has not changed yet - in the conclusions. 

Thank you once again,

Yours sincerely 

the Authors